# Intensive Meadows on Organic Soils of Temperate Climate–Useful Value of Grass Mixtures after the Regeneration

**Teodor Kitczak** [1], **Heidi Jänicke** [2], **Marek Bury** [3] and **Grzegorz Jarnuszewski** [1,*]

1. Department of Environmental Management, West Pomeranian University of Technology in Szczecin, Słowackiego 17 Street, 71434 Szczecin, Poland
2. Landesforschungsanstalt für Landwirtschaft und Fischerei MV, 18196 Dummerstorf, Germany; h.jaenicke@lfa.mvnet.de
3. Department of Agroengineering, West Pomeranian University of Technology in Szczecin, Słowackiego 17 Street, 71434 Szczecin, Poland
* Correspondence: grzegorz.jarnuszewski@zut.edu.pl; Tel.: +48-91-449-6410

**Abstract:** Meadows on organic soils perform an essential role as a source of fodder and biomass for energy purposes. In the case of intensive meadows, it is crucial to use grass mixtures that enable obtaining high yields of good quality; however, on organic soils, the grass species composition changes dynamically. We carried out the full cultivation (ploughing) for grassland restoration. The floristic composition of meadow sward in the first year of full use (2013) was similar to the composition of mixtures used for sowing (2012) individual plots. *Festuca arundinacea* and *Phleum pratense* showed greater resistance to low temperatures in winter and excess water in spring compared to *Lolium perenne*. In comparison, we obtained the highest yield (fresh and dry mass) from mixtures with *Festuca arundinacea* and *Lolium perenne*. We obtained the best quality forage from the first swath and the object with the highest share of *Lolium perenne*. For grassland restoration in the analysed habitat, it is reasonable to use grass mixtures with varied compositions, in which the share of *Lolium perenne* is between 25–50%.

**Keywords:** grass mixtures; weather conditions; organic soils; floristic composition; yields; yield quality

## 1. Introduction

Permanent meadows and pastures cover about 70% of the world's agricultural land and perform a key role in feeding ruminants and other herbivores that produce meat, milk, and dairy products [1]. Changes in diet and population growth increase the demand for animal products and, thus, feed [2]. In the EU, permanent grasslands cover about 34% of agricultural area and are part of mixed cropland systems. In addition to the essential functions related to feed production, permanent meadows perform many functions related to ensuring biodiversity, ecosystem and landscape services, and biomass production for energy [1,3]. Among the ecosystem functions, we should emphasise the role of permanent grasslands on organic, which are highly valued due to their carbon sequestration, hydrological and erosion regulation, nitrogen removal, invasion regulation and livestock number [4]. In the European landscape, grasslands in temperate regions are typical in valley areas with a significant share of organic soils susceptible to anthropogenic pressure, such as drainage and climate changes. Permanent grasslands on drained organic soils are traditionally managed to maximise biomass production [1,5]. A fundamental issue is maintaining the primary (production) and ecosystem function of meadows despite the pressures and changes in climate conditions. Functions related to water retention and purifications, carbon sequestration, and preservation of habitats can only be sustainable if meadows are on organic soils and soils are appropriately maintained (permanent plant cover and turf process) [6]. Grasslands are essential global soil organic matter storage

(SOC), containing 12% of the earth's SOM. SOM averages 331 Mg/ha in temperate climate grasslands and is the third largest global carbon store in soils and vegetation. Appropriate grassland management of meadows, such as increasing forage production, can increase soil carbon [7]. The need to maintain meadows as a resource of biomass and other vital functions while ensuring fodder production at an appropriate level is indisputable. The quality and yield level obtained from grasslands depends on the sward's floristic composition, fertilization, and climate conditions [8,9]. Generally, in a temperate climate, we can distinguish semi-natural grasslands with high levels of biodiversity and grasslands for intensive biomass production with high fertilization levels. However, high biomass production decreases biodiversity [9,10]. Research confirms the positive relationship between the increase in biodiversity and yield quality. A more diverse community has a higher probability of including highly productive species, and differences in species resource acquisition in space and time allow complete utilisation of resources [9,11].

The aim of agricultural grasslands is efficient production of fodder biomass [9,11]. In addition to the biomass yield, yield quality, assessed based on of crude protein, fibre, carbohydrates, fat, or energy content, are also essential in animal nutrition [3,9,12]. Species composition of permanent grasslands changes over time, and yield and quality decrease during ageing; so, periodically, production grasslands require regeneration [13]. Intensive agriculture on grasslands causes restrictions on regeneration. Agricultural practices allowing grass species transport between areas do not continue, and seed sources and species pools are impoverished. [14]. The renovation of grasslands is often considered to increase yield and quality [15], although not all studies confirm this thesis [13]. Grass communities, especially those used by mowing on peat-muck soils are characterised by a stable species composition only for a short time, after establishment or under sowing [13], e.g., tall grasses (*Festuca pratense*, *Phleum pratense*, *Dactylis glomerata*) give way first and are replaced by *Poa pratensis*. The appropriate selection of grass species in mixtures should be determined to maintain the quality and yield of grasslands on organic soils [13,16], where renovation of intensive meadows is required frequently (even every five years) [17]. The suitability of biomass for conservation processes is essential to reduce weather conditions' impact on fodder quality. Currently, the production of haylage and silage increased, while the production of hay is declining. With expected climate changes and the increase in extreme events, such as droughts and floods, forages with wide adaptability to support intensive grassland systems are required [16,18,19]. Therefore, the interest in grass species with high carbohydrate content is growing, hence the growing interest in Braun's fescue —*Festulolium braunii* (K. Richt.) A. Camus and new cultivars of perennial ryegrass (*Lolium perenne*). These species are a valuable component of mixtures for permanent grassland renovation. Perennial ryegrass is Europe's most popular forage grass species, distinguished by its high regrowth capacity, rapid establishment, tolerance to frequent mowing and high nutritive value and digestibility for ruminants [18,19]. *Festulolium braunii* has high yield potential. Good digestibility, favourable protein content, and soluble carbohydrates characterise the feed produced from it [18–20]. The distinguishing feature of the interspecies hybrid is good winter hardiness and resistance to periodic droughts [18,19,21,22]. *Festuca arundinacea* is commonly used in agricultural grasslands in temperate humid climates. This highly competitive species tolerates temporary water logging, is suitable for organic soils, and shows persistence under drought and cutting [19,23,24]. A species with outstanding value and lower tolerance to frequent cutting is *Phleum pratense*, which reduces the risk of yield losses caused by stress factors, e.g., weather conditions [17,19]. *Phleum pratense* has low soil requirements; however, the shallow root system of this species makes it susceptible to drought [25].

The research aimed to evaluate grass mixtures' with *Lolium perenne*, *Festuca arundinacea* and *Phleum pratense* suitability for the regeneration of intensive grasslands on organic soils and their durability in a 6-year experiment in variable weather conditions. The hypothesis assumed that the mixtures with the largest share of *Lolium perenne* would be characterised by the best performance and durability on organic soils.

## 2. Materials and Methods

### 2.1. Study Site and Experiment Design

The research was carried out on grasslands on organic soil of the muck soil type. This area is located in the Randow River valley, near Retzin village (NE Germany—Mecklenburg-Vorpommern) (Figure 1) and is filled with a fluviogenic bog with thickness ranging from 90 cm to 220 cm. The surface layer was rebuilt due to sandblasting and is a mineral-organic formation with a 30–50 cm thickness. The created surface layer has the features of muck soils and is not very susceptible to transformations, and is easy to cultivate in the field.

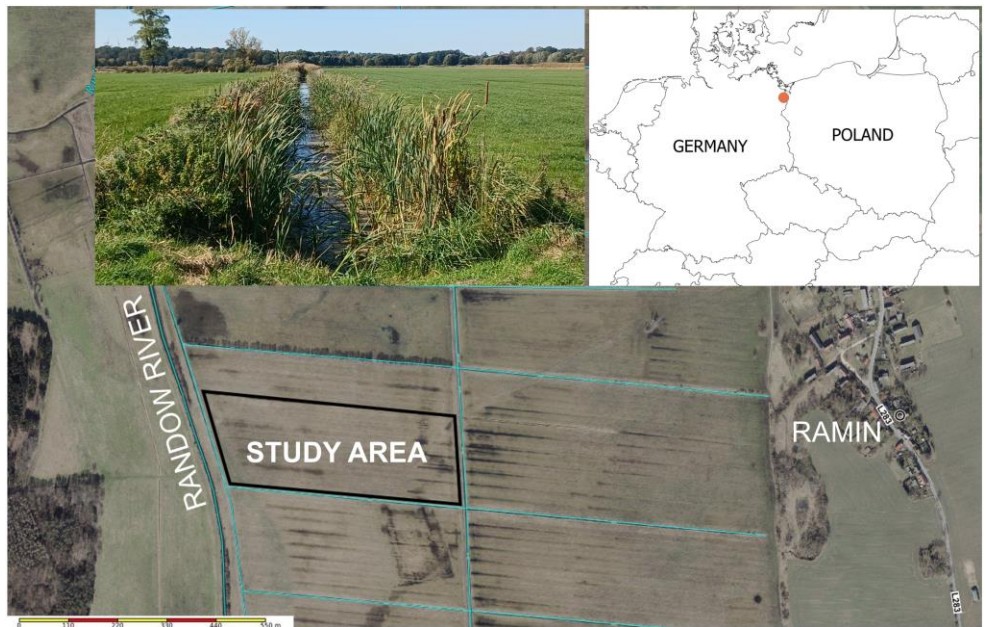

**Figure 1.** Location of study area.

The farm owns the experimental grassland area—Raminer Agrar GmbH & Co KG (Ramin, Germany). The restoration of the grassland was made using the full plough cultivation method. The results presented in this study come from a single-factor experiment from 2013–2018. The experiment was established by the randomised block method in four replications. The single plot area was 10 m². The following species of grasses were used for the regeneration of grasslands: reed fescue (*Festuca arundinacea*), perennial ryegrass (*Lolium perenne*)—a mixture of four cultivars with a share of 25% each: Arusi, Citius, Navarra, and Turandot, timothy meadow (*Phleum pratense*). Table 1 presents the grass mixtures used to sow individual plots of the experiment.

**Table 1.** The share of species in the mixtures %.

| Species and Cultivar | Mixture (Object) [1] | | |
|---|---|---|---|
| | 1 | 2 | 3 |
| *Festuca arundinacea* | 85 | 55 | |
| *Lolium perenne* | 15 | 25 | 100 |
| *Phleum pratense* | - | 20 | |

[1] The percentage share of a given species in the mixture was calculated in relation to the seeding of this species in pure sowing.

The procedures performed each year included: dragging, rolling, fertilisation, and mowing. During the spring dragging and rolling, a multi-component fertiliser was applied—NPK (Mg S) 5-16-24 (4–7), in which the following doses were added: 15 kg N·ha$^{-1}$, 21 kg P·ha$^{-1}$, 60 kg K·ha$^{-1}$, 7 kg Mg·ha$^{-1}$, and 21 kg S·ha$^{-1}$. Additionally, 72 kg N·ha$^{-1}$

was used as an ammonium nitrate and urea (AHL) solution. For the second cut, AHL at a dose of 65 kg N·ha$^{-1}$, and for the third cut—36 kg N·ha$^{-1}$ was used. During the growing seasons, three to five cuts were collected, and their number is given in the tables with the yield of the tested mixtures (Table 2). The meadow's first cut was collected during the stem shooting phase/the beginning of the earing of the dominant species. The subsequent cuts were made at intervals of 4 to 7 weeks. Harvested biomass from the analysed grasslands were allocated to haylage.

**Table 2.** Floristic composition of meadow sward (%) from the first cut in the years of research.

| Object | Species | Share in Mixture (%) | Share on Sward (%) in Years | | | | | |
|---|---|---|---|---|---|---|---|---|
| | | | 2013 | 2014 | 2015 | 2016 | 2017 | 2018 |
| 1 | *Festuca arundinacea* | 85 | 66 | 72 | 68 | 93 | 82 | 65 |
| | *Lolium perenne* | 15 | 34 | 26 | 31 | 5 | 16 | 21 |
| | *Other* | | - | - | 1 | 2 | 2 | 14 |
| 2 | *Festuca arundinacea* | 55 | 62 | 68 | 65 | 72 | 81 | 62 |
| | *Lolium perenne* | 25 | 27 | 25 | 24 | 12 | 10 | 14 |
| | *Phleum pratense* | 20 | 11 | 7 | 9 | 8 | 5 | 8 |
| | *Other* | | - | - | 2 | 8 | 4 | 16 |
| 3 | *Lolium perenne* | 100 | 100 | 100 | 97 | 91 | 90 | 74 |
| | *Other* | | - | - | 3 | 9 | 10 | 26 |

*2.2. Biomass Properties Analysis*

Detailed studies included: analysis of the floristic composition of individual swards, fresh and dry matter yields, the content of crude protein, crude fibre and water-soluble sugars, and net energy lactation value (NET). The content of dry matter, crude proteins, crude fibres, crude fats, and water-soluble sugars was determined in accordance with the methods contained in the VDLUFA method book—the chemical analysis of animal feed [26]. Dry matter was determined using the drying-weigh method with pre-drying. The crude protein content was performed based on Kjeldahl nitrogen by the VDLUFA 3.1.1 method. This method consists of digestion with sulfuric acid in the presence of a catalyst, then the acidic solution is made alkaline with caustic soda, and the released ammonia is distilled into a matrix containing some sulfuric acid, the excess of which is titrated with standard sodium hydroxide solution. The crude fibre was determined in accordance with the VDLUFA 6.1.1 method, which enables the determination of the content of non-fat organic matter insoluble in acids and bases in feed, conventionally expressed as crude fibre. The method consists of degreasing the sample and treating it successively with boiling sulfuric acid and boiling potassium hydroxide of specified concentrations. The residue is separated by filtration through a sintered glass filter, washed, dried, weighed and ashed at 475 °C to 500 °C. The weight loss occurring during combustion corresponds to the crude fibre content of the sample. The crude fat was determined in accordance with the VDLUFA 5.1.1 method, which consists of extracting fats from the sample with petroleum ether; the obtained solvent is distilled off, and the residue is dried and weighed. Soluble sugar was determined with the VDLUFA 7.1.3 method. Water-soluble sugars were extracted with aqueous ethanol, then determined and assayed based on a modified LUFF-SCHOORL method in which copper salts were reduced with inverted sugar using Fehling's solution. The precipitated Cu(I) oxide was separated and weighed. Lactation energy (NET) was calculated based on the equation: NET = 0.6 [1 + 0.004 ($q$ − 57)] × ME were $q$ = ME/GE × 100. ME is the metabolic energy calculated for mixed feed based on the content of ash, crude protein, crude fat, starch, and gas formation. GE is gross energy calculated based on content of crude proteins, crude fats, crude fibre, and N-free extractives. Plant material samples were taken from each plot of 1 m$^2$. To determine the botanical

composition of the cuts, the botanical-weight method was used [27]. The work presents the botanical composition of the first cut, similar to the remaining cuts. Chemical analyses were performed in cooperation with the National Research Institute for Agriculture and Fisheries of Mecklenburg-Vorpommern (Landesforschungsanstalt für Landwirtschaft und Fischerei MV) in Dummerstorf, in the accredited laboratory LUFA Rostock.

### 2.3. Statistical Analysis

The statistical analysis of the results was performed using the ANAWAR 5.3 software (developed by Professor Franciszek Rudnicki) dedicated to agricultural experiments. This software was used to analyse variance with the regression of source Software ANAWAR 5.3 contains computational programs for orthogonal data from single and multiple single-, double- and three-factor experiments. The Tukey test at the level of $p \leq 0.05$ was used to determine the significance of results diversity. The multivariate principal component analysis was performed using Software Statistica 12.5.

### 2.4. Meteorological Condition and Groundwater Level

The meteorological data came from Löcknitz city (Germany) meteorological station. For the analysis of climatic conditions, data from the years 2013–2018 were used, which were characterised against the background of multi-year average 1980–2010 (Figures 2 and 3), indicating a considerable variation in air temperature and the sum of precipitation. During the research, a large fluctuation in the sum of precipitation and temperature were found. The average air temperature and precipitation in the entire growing season in the study years were higher than in the corresponding period in the multi-year period (Figure 3).

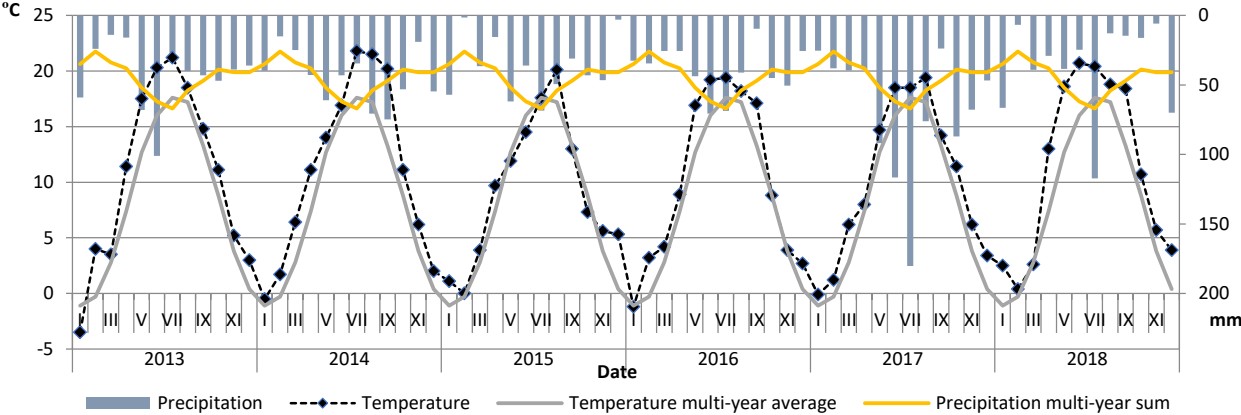

**Figure 2.** Average monthly air temperature and monthly total rainfall in years 2013–2018.

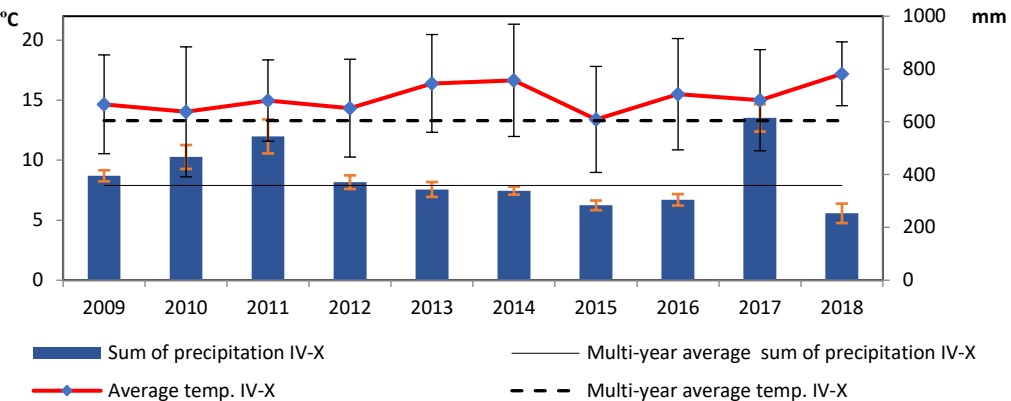

**Figure 3.** Average air temperature and total rainfall (with Standard Deviation) during the growing season in years 2009–2018 and multi-year average air temperature and multi-year average sum of precipitation in the study area.

In all years of the research period, a higher average temperature during the growing season (April–October) was found against multi-year average data. The warmest years were 2018 (17.2 °C), 2014 (16.7 °C) and 2013 (16.4 °C). The coldest year was 2015, with an average temperature of 0.1 °C higher than the multi-year average. The mean values for the growing season exceeded the average value from the multiannual period (13.3 °C) by more than 2.4 °C. The warmest were the summer months from June to September 2013–2017. Exceptional was 2018, in which high average temperatures were recorded throughout the whole growing season concerning to growing season in the multi-year period 1980–2010.

In the years of the research, the amount of precipitation much higher than in many years was recorded in 2017 (Figures 2 and 3). The total rainfall of the growing season of the year 2017 was higher by 256 mm than the average sum of precipitation in the multi-year period. The highest monthly rainfall (180.3 mm) was recorded in July 2017. In other years, rainfall totals during the growing season were similar to the amounts recorded in the multi-year period (359 mm).

The average groundwater level in the study period's growing season (April–October) was 57 cm below the ground level. During the research period, water was recorded on the meadow's surface in the 2017 growing season, while the lowest average groundwater level (88 cm below ground level) occurred in 2018. the observation period (2007–2021) indicates a clear trend of lowering the groundwater level in subsequent years in the analysed area. In addition, increasing water level fluctuations during the growing season have been observed in recent years (Figure 4). Despite the existing infrastructure on the analysed complex of meadows (drainage ditches and gates), it was not possible to avoid lowering the groundwater level caused mainly by higher air temperatures, which increase water losses as a result of evapotranspiration [28,29].

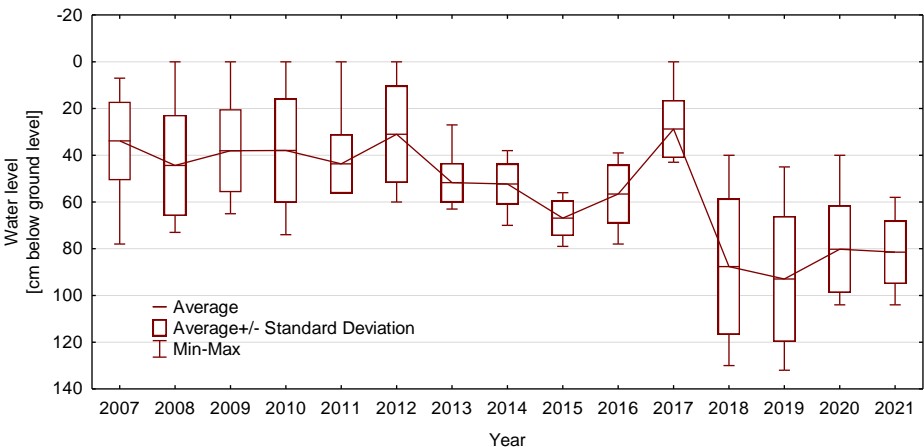

**Figure 4.** Characteristics of the groundwater level in the study area in 2007–2021.

## 3. Results and Discussion

### 3.1. Floristic Composition of Mixtures

The floristic composition of meadow sward in 2013 (the first year of full use) corresponded to the share of individual species used in the mixtures for the regeneration (Table 2). The paper presents only the results concerning the first cut's floristic composition, as they were similar in the remaining cuts. Obtained results showed that the botanical composition in the first three years of the study was stable when analysing the studied mixtures in the years with different weather conditions (precipitation and low temperature drops in spring 2016). An important factor that influenced the composition of the meadow sward was the occurrence of low temperatures in the spring of 2016 which caused the freezing of plants, mainly perennial ryegrass. It resulted in a noticeable reduction in the share of *Lolium perenne* in all grass mixtures (Table 2). A significant decrease in the share of *Lolium perenne* (16%) in the case of mixture 3 was also recorded in 2018 after a very wet year in 2017. This phenomenon has not been observed in *Festuca arundinacea* and *Phleum*

*pratense*. -The share of *Festuca arundinacea* in the meadow swards (where mixtures 1 and 2 were used) remained high (above 60%). In the case of mixture 1, the share of this species in the sward during the research period decreased by 20%, and in the case of mixture 2, it increased by 7% compared to 2013. *Festuca arundinacea* showed high resistance to low temperatures in 2016 and responded with an increase in the percentage of sward by 15% for mixture 1 and 7% for mixture 2, respectively Despite greater resistance of *Phleum pratense* to extreme weather conditions (low temperatures, high humidity, and flooding), compared to *Lolium perenne*, it turned out to be less resistant to intensive use (cutting). This species showed lower competitiveness than others, as evidenced by a significant decrease in the share (9%) already in the second year of full use (Table 2). Becker et al. [18] also confirmed the decrease in the share of *Phleum pratense* in the sward and its lower yield with intensive use (mowing, nitrogen fertilisation) on organic soils. However, the authors indicate that this species is able to compete with *Lolium perenne* on mineral soils.

The participation of *Lolium perenne* in sward regeneration *i* lasted until mid-June. The species composition found in the second cut remained constant in subsequent cuts. As indicated by the results of Kitczak et al. [30] on organic soils, *Festulolium braunii* performs better (maintaining the share in the sward at 52 to 100% after 6 years) on organics soils than *Lolium perenne*. *Festulolium braunii* is characterised by high durability in the sward and significantly less sensitivity to unfavourable weather conditions compared to *Lolium perenne* and about 25–50% inferior to species, such as *Festuca arundinacea* and *Phleum pratense*, which the literature reports as persistent and habitat-developing species well on organic soils [16,25]. Kulik and Baryła [31] confirmed the high durability of the *Lolium perenne* in pasture sward used in the four-species mixture for regeneration. After 12 years with an initial share of 35%, the share of this species in the sward was, on average, 19.9% in the first regrowth and 22.0% in the third regrowth. Despite this, the authors found large fluctuations in the share of *Lolium perenne* in particular years (the share in the sward from 7.3% to 63.0%), which was related to the climate conditions, mainly temperature and humidity. In the conditions of low temperatures in winter and excess water in spring, there was a significant reduction in the share of *Lolium perenne*.

The authors found that perennial ryegrass fell out in cold winters and regenerated in the years when it increased its share in pasture sward. The research did not confirm the rapid regeneration of *Lolium perenne*, as indicated by Kulik and Baryła [31], because it needs water and nitrogen. At the same time, higher average temperatures and lower rainfall characterised the growing season in 2016 compared to the multi-year average. A visible increase in the share of other species characteristic of these habitat conditions (*Poa pratensis*, *Poa trivialis* L., and *Alopecurus pratensis*) was noted after the fourth year of use. It was related to the replacement of *Lolium perenne* in sward by more expansive species [16,17,30,31]. Research shows that *Lolium perenne*, in addition to instability in sward associated with displacement by other species [11,16,18,19], is also sensitive to water deficit [18,19,24]. The observed climatic condition in the study area indicates a deepening water deficit related to the increase in temperatures and the lowering groundwater level (Figures 2–4). On the other hand, meadows on organic soils may periodically experience high groundwater levels (the year 2017). *Festuca arundinacea* is more resistant than *Lolium perenne* and *Festulolium braunii* to extreme climatic conditions (drought and high moisture) due to its deeper root system, but it is characterised by lower digestibility [18,25]. It can be assumed that the ideal solution would be to use a *Festulolium* hybrid that combines the resistance of *Festuca arundinacea* and the high fodder quality of *Lolium perenne* [18–21]. However, a study by Kitczak et al. [30] indicate that this species fell out on organic soils in 2016 due to freezing or excess moisture in 2017. Therefore, according to Østrem et al. [21] *Festulolium* should be a component of a multi-species mixture in unfavourable conditions. The conducted research also confirms the greater usefulness of grass mixtures compared to the monoculture (*Lolium perenne*) [11,32], which had the highest weed infestation in the last year of the experiment.

### 3.2. Dry Matter Yield

The obtained dry matter yields (Table 3) indicate that, on organic soils of river valley areas in temperate climate conditions, multi-species mixtures with a high share of *Festuca arundinacea* (Figure 5) produced the highest yield. The average annual yield of dry matter from mixture 1 (*Festuca arundinacea*—85% and *Lolium perenne* 15%) was 17% higher than the yield obtained from the monoculture of *Lolium perenne* (mixture 3). Similarly, multicomponent mixture 2 was characterised by an average annual yield of 12% higher. Average annual yields in the range 11.5–17.0 t·ha$^{-1}$ and similar relationships-the highest yield from the mixture with the largest share of *Festuca arundinacea* compared to the monoculture *Lolium perenne* and mixtures of the two mentioned was obtained by Cougnon et al. [24] on loam sandy soil. Lower yields of dry matter of mixtures (<10.5 t·ha$^{-1}$) with *Lolium perenne* and *Festuca arundinacea* on peat soils were obtained by Becker et al. [18]. Additionally, lower persistence compared to mixtures on mineral soils characterised mixtures on peat soils. The results of the six years of own research and Kitczak et al. [30] in the same habitat conditions indicate that mixtures of short-lived species (*Lolium perenne* and *Festulolium braunii*) in large share in sward increase dry matter yields in the first four years of intensive use of meadows. In the following years, decreases in average yields were recorded by 34.1%in the fifth and 35.9% in the sixth year of the study (Table 3). The research of Halling [19], Østrem et al. [21], and Kemešytė et al. [23] show that with the increase in the cultivars *Festulolium braunii* and *Lolium perenne*, their yields decreased, which is related to the ageing effect of grassland cultivation. Generally, the ageing process concerned all mixtures, although the weather conditions, e.g., the very wet year in 2017. The highest yield was achieved on all objects in the second year of cultivation. It is related to the effective functioning of the turf's root system in the first years of cultivation after regeneration [13]. The PCA analysis indicates (Figure 5) that the yield depended on water availability (precipitation and groundwater level, which were strongly correlated during the plant growth period). The positive effect of water availability on biomass yield was confirmed on mineral and organic soils [16–19,28,29,31,32].

**Table 3.** Dry matter yield in t·ha$^{-1}$.

| Mixture | Cut | Years | | | | | | Mean |
|---|---|---|---|---|---|---|---|---|
| | | **2013** | **2014** | **2015** | **2016** | **2017** | **2018** | |
| 1 | 1 | 3.78 | 6.11 | 3.93 | 3.44 | 3.70 | 4.33 | 4.22 |
| | 2 | 4.18 | 4.40 | 3.24 | 3.30 | 2.57 | 2.25 | 3.32 |
| | 3 | 6.22 | 5.31 | 4.15 | 5.22 | 7.21 | 4.49 | 5.43 |
| | 4 | 2.72 | 4.14 | 3.09 | 2.95 | n. d. * | 3.72 | 3.32 |
| | 5 | n. d. * | 2.83 | 4.05 | 3.10 | n. d. * | n. d. * | 3.33 |
| | Total | 16.9 | 22.79 | 18.46 | 18.01 | 13.48 | 14.79 | 17.41 |
| 2 | 1 | 3.78 | 6.80 | 4.24 | 3.37 | 3.84 | 3.87 | 4.32 |
| | 2 | 4.49 | 4.09 | 2.90 | 3.00 | 2.58 | 2.07 | 3.19 |
| | 3 | 5.20 | 4.98 | 3.78 | 5.66 | 5.29 | 3.78 | 4.78 |
| | 4 | 3.37 | 4.08 | 3.15 | 3.18 | n. d. * | 3.08 | 3.37 |
| | 5 | n. d. * | 2.58 | 3.69 | 2.76 | n. d. * | n. d. * | 3.01 |
| | Total | 16.84 | 22.53 | 17.76 | 17.97 | 11.71 | 12.80 | 16.60 |
| 3 | 1 | 3.25 | 4.93 | 2.43 | 1.24 | 3.73 | 3.84 | 3.24 |
| | 2 | 4.99 | 3.24 | 4.22 | 4.03 | 2.96 | 2.09 | 3.59 |
| | 3 | 4.56 | 3.83 | 3.89 | 5.81 | 3.40 | 3.27 | 4.13 |
| | 4 | 2.60 | 3.64 | 3.10 | 2.69 | n. d. * | 1.94 | 2.79 |
| | 5 | n. d. * | 2.68 | 3.71 | 3.00 | n. d. * | n. d. * | 3.13 |
| | Total | 15.40 | 18.32 | 17.35 | 16.77 | 10.09 | 11.14 | 14.85 |
| Mean of years | | 16.38 | 21.21 | 17.86 | 17.58 | 11.76 | 12.91 | 16.28 |
| HSD 0.05 | | i. d. ** | 1.95 | 1.06 | i. d. ** | 2.73 | 3.05 | 0.65 |

* no data, ** insignificant differences.

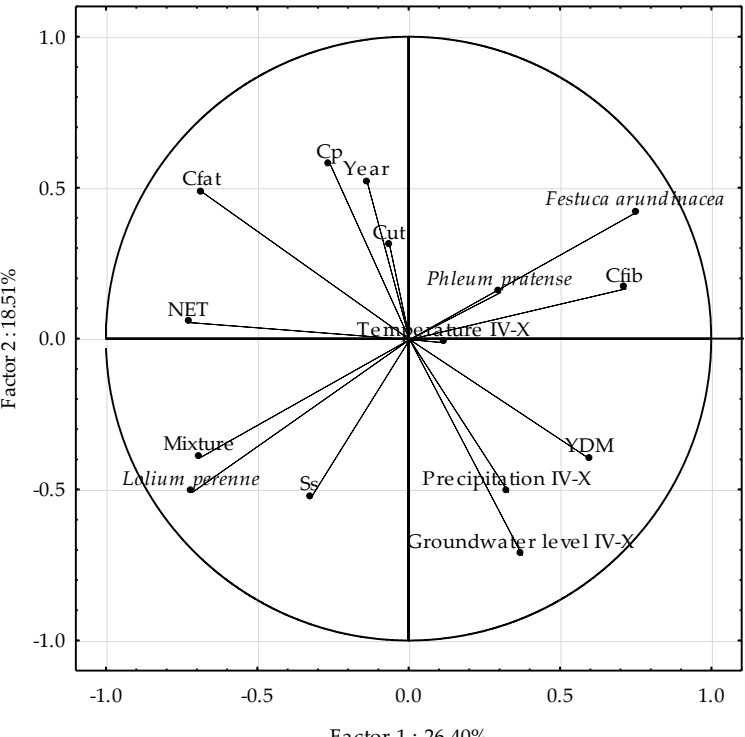

**Figure 5.** The principal component analysis (PCA) for mixtures, biomass yield, forage value, and weather conditions. YDM—dry matter yield, NET—net energy lactation, Cp—crude protein, Cfat—crude fat, Cfib–crude fibre, Ss—soluble sugars.

### 3.3. Fodder Quality

The harvested crop can assess the forage per unit area, but its quality is an equally important factor. The research confirms other results [16–18,21,25], which showed that the chemical composition and energy concentration depend on the sward's floristic composition. Other studies of grass mixtures used for grassland regeneration indicate the highest crude protein content in the sward from the multi-component mixtures with *Festulolium braunii*, *Lolium perenne*, *Poa pratensis*, and *Phleum pratense* [30,33]. This research (Table 4) shows that in the years and object when the share of *Lolium perenne* in individual objects' floristic composition was significant, the forage value of the sward was the best, especially in terms of soluble sugar content (Figure 5). The best nutritional values, taking into high crude protein content, net energy concentration, and the lowest crude fibre content, were characterised by biomass obtained from the fourth and fifth cuts.

Overall, the protein content in the research objects ranged from 144.3 to 201.3 g·kg$^{-1}$ DM (Table 4). The highest content of crude protein was found in mixtures 3 (*Lolium perenne*), averaging 183.5 g·kg$^{-1}$ which is in line with the opinion of other authors about the high nutritional value of these species [18–22,30,33]. The lowest average crude protein content occurred in mixture 1, with the highest share of *Festuca arundinacea*.

Crude protein content in the tested mixtures was higher than in the research of Olszewska [12] for *Lolium perenne* on Haplic Cambisol (Eutric) from clay and Cougnon et al. [24] for *Festuca arundinacea* and *Lolium perenne* on sandy loam soil at and similar to the results for mixtures of similar composition on organic soils [30,33]. The crude protein content increases between the first and fifth harvests, as in studies by Olszewska [12] and Czyż et al. [33], which relates to the transition from generative to vegetative growth [34]. Regarding the content of soluble sugars (68.0–135.5 g·kg$^{-1}$ DM), the sward from site 1 stood out, with a floristic composition, which included only two species (*Festuca arundinacea* and *Lolium perenne*). It is consistent with the opinion of Curran et al. [20], Kitczak et al. [30], and Downing and Gamroth [35], who claim that their high soluble sugar content distinguishes *Lolium perenne* and *Festulolium braunii* from other species.

**Table 4.** The content of organic compounds (g·kg$^{-1}$ DM) and the concentration of net energy (MJ·kg$^{-1}$DM), the average of the years 2013 to 2018.

| Mixture | Cut | Crude Protein | Crude Fibre | Crude Fat | Soluble Sugars | The Net Energy |
|---|---|---|---|---|---|---|
| 1 | 1 | 152.2 | 281.3 | 20.2 | 92.0 | 5.9 |
| | 2 | 148.7 | 280.8 | 21.0 | 95.5 | 6.0 |
| | 3 | 145.2 | 272.7 | 22.2 | 122.7 | 6.1 |
| | 4 | 145.5 | 255.5 | 22.8 | 131.0 | 6.3 |
| | 5 | 144.3 | 256.2 | 23.7 | 135.5 | 6.4 |
| | Mean | 147.2 | 269.3 | 22.0 | 115.3 | 6.1 |
| 2 | 1 | 167.5 | 290.2 | 20.8 | 74.2 | 6.0 |
| | 2 | 166.7 | 284.5 | 21.2 | 77.7 | 6.0 |
| | 3 | 153.0 | 281.0 | 21.7 | 90.3 | 6.0 |
| | 4 | 161.2 | 273.8 | 23.2 | 90.3 | 6.1 |
| | 5 | 153.5 | 292.5 | 21.8 | 89.8 | 5.9 |
| | Mean | 160.4 | 284.4 | 21.7 | 84.5 | 6.0 |
| 3 | 1 | 171.3 | 295.3 | 23.0 | 68.0 | 6.0 |
| | 2 | 180.3 | 284.3 | 22.7 | 75.0 | 6.1 |
| | 3 | 178.3 | 280.0 | 23.3 | 70.7 | 6.2 |
| | 4 | 201.3 | 265.3 | 21.3 | 72.3 | 6.2 |
| | 5 | 186.3 | 265.3 | 23.3 | 73.0 | 6.1 |
| | Mean | 183.5 | 278.1 | 22.7 | 71.8 | 6.1 |
| Mean for cuts | 1 | 163.7 | 288.9 | 21.3 | 78.1 | 6.0 |
| | 2 | 165.2 | 283.2 | 21.6 | 82.7 | 6.0 |
| | 3 | 158.8 | 277.9 | 22.4 | 94.6 | 6.1 |
| | 4 | 169.3 | 264.9 | 22.4 | 97.9 | 6.2 |
| | 5 | 161.4 | 271.3 | 22.9 | 99.4 | 6.1 |
| Mean | | 163.7 | 277.3 | 22.1 | 90.5 | 6.1 |

The highest soluble sugar concentration was recorded in the fifth cut (99.4 g·kg$^{-1}$ DM) and the lowest in the first (78.1 g·kg$^{-1}$ DM). At the same time, other authors [12,33,36–38] analysing mixtures with *Lolium perenne* and *Festulolium braunii* obtained the highest concentration of sugars in the first cuts. Large fluctuations in the soluble sugar content may also be related to the weather conditions and water availability [12]. It is related to using carbohydrates for protein production and more intensive cellular respiration during the summer. Other studies [34,36] also confirmed the decrease in the content of sugars with the increase in temperature. The effect is also related to the increasing share of crude fibre and crude protein in subsequent cuts, which also affected the energy content, significantly the share of which was significantly correlated with soluble sugar and crude fat contents (Figure 5). Similar values characterised the concentration of sward energy from all objects: 5.9 to 6.4 MJ·kg$^{-1}$ NET DM. When assessing the meadow sward's fodder value based on the net energy concentration, it was at the level recommended for fodder (6 MJ·kg$^{-1}$ DM), which ensures the proper development of farm animals [30,33,36].

The crude fibre content was within the 255.5—-295.3 g·kg$^{-1}$ DM range. The upper average value (284.4 g·kg$^{-1}$ DM) relates to site 2. Too high a fibre content worsens forage digestibility [30,35,36]. The crude fibre content decreased with successive cuts but remained at a high level, close to recommended or higher (<280 g·kg$^{-1}$ DM) in the feed for ruminants [38].

In terms of parameters determining the quality of the crop in terms of weather conditions, the content of protein and fats was negatively dependent on the availability of water (Figure 5). It was probably related to the increase in the share of *Festuca arundinacea* in the mixtures (1 and 2), which coped best in the conditions of periodic flooding (2017) or low temperatures (2016) and which shows the worst qualitative properties of the fodder.

## 4. Conclusions

The results of floristic composition show that the habitat conditions were favourable for the emergence and further development of plants of the grass species used in the mixtures, as evidenced by the high stability of the botanical composition of the meadow sward with the participation of species in mixtures used for sowing particular objects. Snow-free winter and a significant drop in temperature in early spring in 2016 caused *Lolium perenne* plants to freeze over. *Festuca arundinacea*, and *Phleum pratense* proved to be resistant to these conditions. The plant sown with a mixture of the composition—*Festuca arundinacea*—85% + *Lolium perenne*—15% was characterised by the highest production potential of dry matter; the lower average yield from the research years by 4.7% was obtained from objects sown with the *Festuca arundinacea* mixture—55% + *Lolium perenne*—25% + *Phleum pratense*—20% On the other hand, the dry matter yields of mixtures containing four *Lolium perenne* spp. on average, cultivars were 14.7% lower compared to the *Festuca arundinacea* + *Lolium perenne* mixture, with their regress being more significant in the following years of use. The research hypothesis of the highest quality of mixtures with *Lolium perenne* on organic soils in temperate climate conditions was confirmed. When analysing the parameters of fodder quality (crude protein, soluble sugars, crude fibre, net energy), it should be stated that they were at a similar level at all objects but largely depended on the harvested cut. Mixture 1, with a large proportion of *Festuca arundinacea*, was more resistant to frost and flooding and gave the highest yield, and the share of *Lolium perenne* in this mixture had a positive effect on its quality.

The overall results indicate that in the analysed habitat, it is reasonable to use more grass species for mixtures in the regeneration of grassland regeneration to ensure the stability of yielding and fodder with good quality indicators.

**Author Contributions:** Conceptualization, T.K. and H.J.; methodology, T.K. and H.J.; software, T.K. and G.J.; validation, T.K., H.J. and M.B.; formal analysis, G.J.; investigation, T.K., H.J. and M.B.; resources, T.K. and H.J.; data curation, T.K. and H.J.; writing—original draft preparation, T.K.; writing—review and editing, G.J.; visualisation, T.K. and G.J.; supervision, T.K. and H.J.; project administration, T.K. and H.J.; funding acquisition, T.K. and H.J. All authors have read and agreed to the published version of the manuscript.

**Funding:** This research received no external funding.

**Institutional Review Board Statement:** Not applicable.

**Data Availability Statement:** West Pomeranian University of Technology in Szczecin, Poland.

**Acknowledgments:** Special thanks to Raminer Agrar GmbH & Co KG for successful long-term cooperation.

**Conflicts of Interest:** The authors declare no conflict of interest.

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
