# Peer review of "Intensive Meadows on Organic Soils of Temperate Climate–Useful Value of Grass Mixtures after the Regeneration"

_agriculture, doi:10.3390/agriculture13061126_

Round 1

Reviewer 1 Report

The paper entitled 'Intensive meadows on organic soils...' fits within the scope of the journal. The topic is relevant and the statistical analyses adopted are sound and technically correct. The merit of the paper is high. Further, it considers an area of pasture/grassland on organic soils. However, the paper was difficult to understand in one go. It needed several reads before one could comprehend what the authors are trying to convey. Nevertheless, the manuscript needs a thorough rewrite and the sections that need attention have been highlighted (please see the attached ms).

One issue that is of concern is that the paper on similar lines has been published earlier. Data on DM, the proportion of sward, and quality composition. It would be better if those data are removed before submission. 

Please avoid repetition.

Please check the Table numbering.

References should be reduced by at least 15-20%.

Substantial linguistic improvements are needed in the MS.

Author Response

Thank you for your valuable comments that helped to organize the content and facilitate the reading of the article. All suggestions given in the review have been followed.

A large number of changes have been introduced in the text resulting from the need to obtain an appropriate level of originality of the content as suggested by the editors.

As suggested, the text was linguistically corrected (confirmation in the attachment)

One issue that is of concern is that the paper on similar lines has been published earlier. Data on DM, the proportion of sward, and quality composition. It would be better if those data are removed before submission. 

Answer:

To ensure the originality of the text, the results for mixtures 3 and 4, which were partly published in the previous article, were removed, and corrections were made in the text resulting from these changes.

Please avoid repetition.

Answer:

Suggestions have been followed, changes have been made to the text

Please check the Table numbering.

Answer:

table numbering has been corrected.

References should be reduced by at least 15-20%.

Answer:

9 references were removed, including a duplicate item

Reviewer 2 Report

The present manuscript is interesting and informative and the subject is within scope of the journal. But there exist several shortcomings that should be corrected.

My further comments are included below.

In general, ıntroduction section reflects importance and originally of the study.

Material Method

The materials and methods used in the study were clearly described in the manuscript. Standard methods were given by references.

Line 142. Tables 2 was cited, but it does not exist in text.

Line 187 – 196. This paragraph is the same as above.

Table 3. Although Phleum pratense was resistant to constraint conditions, it should be explained why its ratio in the mixture was decreased over the years.

Results and Discussion

Deeper discussion is needed about floristic composition.

Tables and figures should be cited consecutively order.

Line 308. Replace ‘’Biomass Yield Quality’’ with ‘’ Fodder Quality’’

Author Response

Thank you for the constructive comments that helped to improve the text. All suggestions have been introduced in the text.

The updated text contains many changes related to the need to carry out linguistic corrections and the reviewer's suggestions. In addition, in order to maintain the originality of the article, changes were made as suggested by the editors.

Detailed answers are below.

Line 142. Tables 2 was cited, but it does not exist in text.

Answer: table numbering has been corrected

Line 187 – 196. This paragraph is the same as above.

Answer: repetition removed from text

Table 3. Although Phleum pratense was resistant to constraint conditions, it should be explained why its ratio in the mixture was decreased over the years.

Answer: the reasons for the decreasing share of Phleum pratense in the text were explained

Results and Discussion

Deeper discussion is needed about floristic composition.

Answer: the discussion on changes in the floristic composition was extended

Tables and figures should be cited consecutively order.

Answer: changes have been made to the text as suggested

Line 308. Replace ‘’Biomass Yield Quality’’ with ‘’ Fodder Quality’’

Answer: changes have been made to the text as suggested

Round 2

Reviewer 1 Report

The authors have addressed the points suggested and sufficient improvements in English have been done.

Now the MS is readable and can be published.

Author Response

Thank you for your comments, corrections were made as suggested in review 1.